# Dual Tunable MZIs Stationary-Wave Integrated Fourier Transform Spectrum Detection

**DOI:** 10.3390/s21072352

**Published:** 2021-03-28

**Authors:** Xinyang Chen, Peijian Huang, Ning Wang, Yong Zhu, Jie Zhang

**Affiliations:** The Key Laboratory of Optoelectronic Technology & System, Education Ministry of China, Chongqing University, Chongqing 400044, China; 201908131063@cqu.edu.cn (X.C.); 20113269@cqu.edu.cn (P.H.); ningw@cqu.edu.cn (N.W.); yongzhu@cqu.edu.cn (Y.Z.)

**Keywords:** DTM-SWIFT, optical distribution characteristics, high spectral resolution, Raman spectrum

## Abstract

In order to resolve spectral alias due to under sampling in traditional stationary-wave integrated Fourier transform (SWIFT) spectrometers, an all-on-chip waveguide based on dual tunable Mach-Zehnder interferometer (MZI) stationary-wave integrated Fourier transform technology (DTM-SWIFT) is proposed. Several gold nanowires are asymmetrically positioned at two sides of zero optical path difference and scatter the interference fringes information, which can avoid aliasing of spectral signals and help to gain high spectral resolution. A systematic theoretical analysis is carried on in detail, including the optical distribution characteristics based on multi-beam interference, stationary-wave theorem and signal reconstruction method based on the FT technology. The results show that the method can complete a resolution of 6 nm for Gauss spectrum reconstruction using only 6 gold nanowires, and a resolution of 5 cm^−1^ for Raman spectrum reconstruction using 25 gold nanowires.

## 1. Introduction

The on-chip spectrum detection system plays a vital role in the spectrum instrument, widely used in chemical and biological analysis, environment monitoring, hyperspectral imaging and other fields [1,2,3,4]. Conventional spectrometers are usually bulky and expensive. Integrated technology is an effective way to realize chip-level spectrometers, with advantages of high compactness, high compatibility and low cost [5,6,7,8,9]. Presently, most on-chip spectrometers are based on dispersive elements, such as arrayed waveguide gratings (AWGs) [10,11,12] and planar concave gratings [13,14,15,16], which are very similar to conventional spectrometers with gratings as the core. The realization of on-chip spectrometers can also utilize the characteristics of photonic devices, such as photonic crystals [17,18] and random photonic structures [19]. These on-chip spectrometers can achieve relatively high resolution, but require a large number of detectors to detect large bandwidth light sources, resulting in complex device structures and greatly reducing the signal-to-noise ratio (SNR). The on-chip plasmonic spectrometer can perform spectral sensing and detection through two holographic gratings, and its typical resolution can reach nm level [20,21,22].

Fourier transform spectrometers can effectively solve such problems, achieve high resolution and high signal-to-noise ratio [23,24], such as stationary-wave integrated Fourier transform spectrometers (SWIFT) [25,26,27] and spatial heterodyne spectrometers (SHS) [8,28,29,30]. When the number of MZI used is 32, the reported SHS can achieve a resolution of ~0.045 nm, and the MZI arrays are used to uniformly sample each point in the interferogram. It increases the size and the complexity of the device. A reported SWIFT can achieve a resolution of 4 nm at the center wavelength of 1500 nm [25], using specular reflection to form stationary waves. In addition, in order to satisfy the Nyquist-Shannon criterion in SWIFT, the distance between two detectors must be less than *λ*/4*n* (the wavelength *λ*, the refraction index *n*). This requirement for a typical micro-sized CCD (charge coupled device) is impossible [25,27]. Furthermore, the reconstruction spectral bandwidth (∆σ = 1/(4*n*∆*z*)) is also limited by the sampling step (∆*z*). In recent years, a thermal or electrical tuning method has been used to realize an on-chip integrated FT spectrum detection [31,32], but the system requires an external reflecting mirror and there is a crosstalk effect between neighboring detectors.

In order to complete an all-on-chip method and avoid the crosstalk effect, we propose an all-on-chip Si_3_N_4_ waveguide with dual tunable MZIs. We choose Si_3_N_4_ waveguide because Si_3_N_4_ material has many advantages. First of all, Si_3_N_4_ has a wide band gap (Eg~5.1 eV), and the transparent window can extend from infrared to visible, and even ultraviolet. The second is the high refractive index contrast of Si_3_N_4_, which provides a favorable condition for small-sized and highly integrated single-mode waveguides. In the spectral detection process, we use asymmetrically located gold nanowire arrays to uniformly sample the interference fringes. By changing the phases of two MZIs dynamic arms, an interferogram satisfying the Nyquist-Shannon criterion can be recorded, and the large spectral bandwidth optical sources with a high resolution are reconstructed by Fourier transform technology.

## 2. Theoretical Analysis of DTM-SWIFT

### 2.1. Spectral Detection System

A dual tunable MZIs stationary-wave integrated Fourier transform (DTM-SWIFT) spectrum detection system is shown in Figure 1. The incident spectrum enters two MZIs simultaneously. Outputs *E*_1_ and *E*_2_ come from MZI_1_, and outputs *E*_3_ and *E*_4_ come from MZI_2_. Four light waves have vector superposition on the straight waveguide *L*_5_. The middle fringe has a larger intensity than that of two symmetrically distributed fringes, which is called static fringe; while two symmetrically distributed fringes are called dynamic fringes. Gold nanowire arrays are asymmetrically positioned on both sides of OPD = 0 (optical path difference) to extract the corresponding fringe information, and each corresponding information is recorded by a corresponding detector. Constantly changing the phase difference between two arms leads to the corresponding redistribution of dynamic fringes. After multiple phase modulations, the gold nanowire arrays extract various parts of the dynamic interferograms, also the sampled data are superimposed to form complete interference data, which are used to reconstruct the incident spectrum using Fourier transform technology.

### 2.2. Theoretical Analysis

Shown in Figure 1, we set the initial arm length *L*_1_ = *L*_3_ and *L*_2_ = *L*_4_. The initial length difference ∆*L* is set to ∆*L_Gauss_* and ∆*L_Raman_* to avoid sampling static fringes when the incident light is Gauss signal and Raman signal, respectively. At the propagation direction *z*, the light output from MZI_1_ and MZI_2_ is expressed as:(1a)$$E_{1}=E_{0}e^{i\left(\phi_{1}+\frac{2\pi }{\lambda }n_{5}z\right)}e^{az}$$
(1b)$$E_{2}=E_{0}e^{i\left(\phi_{2}+\frac{2\pi }{\lambda }n_{5}z\right)}e^{az}$$
(1c)$$E_{3}=E_{0}e^{i\left(\phi_{3}-\frac{2\pi }{\lambda }n_{5}\left(z-L_{5}\right)\right)}e^{-a\left(L_{5}-z\right)}$$
(1d)$$E_{4}=E_{0}e^{i\left(\phi_{4}-\frac{2\pi }{\lambda }n_{5}\left(z-L_{5}\right)\right)}e^{-}{}^{a\left(L_{5}-z\right)}$$
where *ϕ*_1_, *ϕ*_2_, *ϕ*_3_, *ϕ*_4_ are the phases of each optical signal output from two MZIs, *ϕ_i_* = 2*πn_i_L_i_*/*λ*, *i* = 1, 2, 3, 4; *E*_0_ is the initial incident amplitude; α is the transmission loss. The total optical field on *L*_5_ is:(2)$$E_{Total}=E_{1}+E_{2}+E_{3}+E_{4}$$

Herein, the refractive index of static waveguides and dynamic waveguides is *n*_2_ = *n*_4_ = *n*_5_ = *n_s_* and *n*_1_ = *n*_3_ = *n_d_*. The phase difference is set *ϕ*_1_ = *ϕ*_3_ and *ϕ*_2_ = *ϕ*_4_. Assuming the transmission loss α = 0, the complex amplitude distribution can be simplified to:(3)$$E_{Total}=E_{0}\left(e^{i\left(\frac{2\pi }{\lambda }n_{s}z+\phi_{1}\right)}+e^{i\left(\frac{2\pi }{\lambda }n_{s}z+\phi_{2}\right)}+e^{i\left(\frac{2\pi }{\lambda }n_{s}\left(L_{5}-z\right)+\phi_{1}\right)}+e^{i\left(\frac{2\pi }{\lambda }n_{s}\left(L_{5}-z\right)+\phi_{2}\right)}\right)$$

Then the intensity is:(4)$$\begin{array}{ll} I=E_{Total}^{*}\times E_{Total}= & E_{0}\left(e^{i\left(\frac{2\pi }{\lambda }n_{s}z+\phi_{1}\right)}+e^{i\left(\frac{2\pi }{\lambda }n_{s}z+\phi_{2}\right)}+e^{i\left(\frac{2\pi }{\lambda }n_{s}\left(L_{5}-z\right)+\phi_{1}\right)}+e^{i\left(\frac{2\pi }{\lambda }n_{s}\left(L_{5}-z\right)+\phi_{2}\right)}\right)\times \\ & E_{0}\left(e^{-i\left(\frac{2\pi }{\lambda }n_{s}z+\phi_{1}\right)}+e^{-i\left(\frac{2\pi }{\lambda }n_{s}z+\phi_{2}\right)}+e^{-i\left(\frac{2\pi }{\lambda }n_{s}\left(L_{5}-z\right)+\phi_{1}\right)}+e^{-i\left(\frac{2\pi }{\lambda }n_{s}\left(L_{5}-z\right)+\phi_{2}\right)}\right) \end{array}$$

According to Equation (4), we can get:(5)$$I=E_{0}^{2}\times \left(\begin{array}{l} 4+2e^{i\left(\phi_{1}-\phi_{2}\right)}+2e^{-i\left(\phi_{1}-\phi_{2}\right)}+2e^{i\left(k\left(2z-L_{5}\right)\right)}+2e^{-i\left(k\left(2z-L_{5}\right)\right)}+e^{i\left(k\left(2z-L_{5}\right)+\left(\phi_{1}-\phi_{2}\right)\right)}\\ +e^{-i\left(k\left(2z-L_{5}\right)+\left(\phi_{1}-\phi_{2}\right)\right)}+e^{i\left(k\left(2z-L_{5}\right)+\left(\phi_{2}-\phi_{1}\right)\right)}+e^{-i\left(k\left(2z-L_{5}\right)+\left(\phi_{2}-\phi_{1}\right)\right)} \end{array}\right)$$

According to Euler’s formula *e^i^*^(*x*)^ = *cos*(*x*) *+ isin*(*x*) and the initial intensity *I*_0_ = *|E*_0_*|*^2^, we can get the intensity at all wavelength *λ*:(6)$$I=E_{Total}^{*}\times E_{Total}=\int_{\lambda_{1}}^{\lambda_{2}}\int_{z_{1}}^{z_{2}}4I_{0}\left[1+cos\left[\frac{2\pi }{\lambda }\times n_{s}\times 2\left(z-\frac{L_{5}}{2}\right)\right]\right]\left[1+cos\left(\phi_{1}-\phi_{2}\right)\right]dzd\lambda$$
where the incident wavelength range is from *λ*_1_
*to*
*λ*_2_. The right side of the equation can be regarded as two parts, one part is related to the distance *z*, similar to the distribution of stationary waves, and the other part is related to the phase difference ∆*ϕ = ϕ*_1_ − *ϕ*_2_ between two arms of each MZI.

Herein, we take Si_3_N_4_ waveguide as an example, ignoring the influence of dispersion. A thermal tuning method is used to realize phase tuning of the dynamic arms, and only the first-order influence of the thermo-optical characteristics is considered. The phase difference can be expressed as:(7)$$\Delta \phi \left(T\right)=\frac{2\pi }{\lambda }\left(\Delta n\left(T\right)L_{1}+n_{s}\Delta L\right)$$
(8)$$\Delta n\left(T\right)=\frac{dn_{Si_{3}N_{4}}}{dT}\Delta T$$
where ∆*n* is the change of refractive index under temperature tuning. The first-order parameter of Si_3_N_4_ thermo-optical coefficient (TOC) at room temperature [33,34] is:(9)$$\frac{dn_{Si_{3}N_{4}}}{dT}\left(T\right)=5.663\times 10^{-5}+1.511\times 10^{-9}T+7.047\times 10^{-11}T^{2}$$

The reconstructed spectral bandwidth is expressed as ∆σ = 1/(4*n*∆*z*), where ∆*z* is one-step movement distance of the fringe during the tuning process, determined by the thermal tuning method. The spectral resolution is expressed as *R* = 1/(2*n_s_z_max_*). In the expression, *z_max_* is the maximal distance that gold nanowire arrays can extract the dynamic interferogram.

## 3. Results

In the process of sampling and detection, there are two important issues to consider. Due to the low light energy of evanescent wave, the use of nanowire arrays (e.g., gold) as sampling equipment is more suitable than that of nano-groove arrays. Gold nanowire arrays are able to both scatter the light and sample the interferogram, and the intensity of scattered light is enhanced compared to that of the nano-groove arrays. Although SNOM (scanning near-field optical microscopy) tips are ideal probes to sample the evanescent wave with nanometre resolution, SNOM is not compatible with the idea of compact integrated and monolithic optical devices.

Therefore, we use CCD as the detection equipment. Le Coarer et al., confirmed in experiments that the signals collected by gold nanowires can be detected in the far field using a simple CCD. We design the pitch of the gold nanowire arrays to reduce the signal crosstalk, and we design the size of the nanowires as small as possible to reduce the damage to the interference pattern and restore it to the greatest extent.

### 3.1. Gauss Spectrum as an Input

Taking Gauss spectrum as an input, we set the incident center wavelength *λ_central_* of 785 nm, the full width at half maximum (FWHM) of 35 nm, *L*_1_ = *L*_3_ = 12 mm, *L*_5_ = 40 mm and the refractive index *n_s_* = 2.25. The phase change of two arms is ∆*ϕ* = 2*π L*_1_∆*n*/*λ*, while ∆*n* is set to 0.003, 0.006, and 0.009. The loss coefficient of Si_3_N_4_ can reach 0.045dB/m under better processing technology. According to the calculation, when the transmission distance is the longest 32 mm, the loss is only 0.03%, so it can be ignored. Even when the loss coefficient increases to 1 dB/m~2 dB/m due to some defects in the process, the loss is only 0.73%~1.46%, so we consider the transmission loss α = 0. According to Equation (6), the optical signal distribution on *L*_5_ is shown in Figure 2.

There are four parameters to be taken consideration: the initial waveguide difference ∆*L*, maximal sampling range *z_max_*, sampling interval ∆*z* and numbers of nanowires.

In order to avoid sampling the static fringes data, the initial phase difference must be chosen carefully, because it determines the distribution of dynamic fringes. Based on Equations (6) and (7), the expression of the initial phase difference is ∆*ϕ* = 2π(*n_s_*∆*L*)/*λ(*∆*n* (*T*) = 0 and the initial refractive index is equivalent), and by calculation we get ∆*z_central_* = 0.5∆*L* ≈ 2.6667 × 10^−3^∆*n*, where ∆*z_central_* is the center of the dynamic fringes position away from OPD = 0. We calculated that when the initial detection position is 6 μm (based on *λ*^2^/∆*λ*) far away from OPD = 0, it is almost impossible to record any static fringe data. Therefore, we set ∆*L* = 32 μm to achieve a dynamic peak position distance of 16 μm away from OPD = 0. We start to detect at a distance of 9 μm from OPD = 0.

According to the resolution *R* = 1*/*(2*n_s_z_max_*), we set *z_max_* = 24 μm in order to achieve a resolution of 6 nm at 785 nm. Based on the Nyquist-Shannon criterion and the ∆σ = 1/(4*n*∆*z*), the sampling interval should be less than 87.2 nm at central wavelength of 785 nm, so the reconstructed spectral bandwidth ∆σ is larger than 1.274 × 10^4^ cm^−1^.

When only one gold nanowire is used, the required number of moving steps is *z_max_*/∆*z*. Taking consideration of a 7 μm pixel size of the traditional detector, the number of gold nanowires and the number of tuning, we set ∆*z* = 40 nm, and six gold nanowires are asymmetrically placed on the *L*_5_. 6 gold nanowires positioned away from OPD = 0 are 13 μm, 21 μm, 29 μm on the left, 17 μm, 25 μm, 33 μm on the right, respectively. Based on Equation (9), the first-order TOC coefficient of Si_3_N_4_ is 6.2 × 10^−5^ K^−1^ at room temperature of 300 K, then ∆*z_central_ ≈* 2.6667 × 10^−3^ ∆*n*
*=* 1.6534 × 10^−^^7^∆*T* and the temperature change ∆*T* is 0.242 K when the sampling interval is 40 nm. The temperature tuning times are 100 and the maximal temperature change is 24.2 K. When the peak shift caused by temperature error exceeds 87.2 nm (*λ*/4*n*), the spectrum will be aliased and part of the spectrum will be lost. At this time, the temperature error is 0.5275 K. For some existing on-chip heating systems, this problem can be avoided, so we ignore its influence in this paper. The data recorded by each nanowire are shown in Figure 3a. The original incident Gauss spectrum and the reconstructed data are shown in Figure 3b, while an enlarged difference is shown in Figure 3c. The position of the gold nanowires corresponding to the dynamic interferogram is shown in Figure 3d.

### 3.2. The Nanowire Width Effect

On the one hand, in order to reconstruct spectral information as ideally as possible, the width of the gold nanowires should be considered and be as small as possible (combined with the current processing technology). On the other hand, according to Equation (6), the total intensities on *L*_5_ are superimposed including the total wavelengths. When the gold nanowire width *w* is exactly an integer multiple of the stationary wave period, *w* = *K* × *(**λ*/2*n_s_)*, where *K* is an integer, *K* = 1, 2, 3*…m*, all sampling data for this wavelength are fixed values, which leads to that the reconstructed information at this wavelength equals zero.

In the process of gold nanowire scattering evanescent waves and CCD detected signals, the evanescent wave is in the near field, and the CCD is in the far field. This is a complex conversion process, with the influence of scattering-diffraction. For simple calculation, we adopt a normalization method to approximate the influence of scattering-diffraction, and use an integral method to deal with the influence of nanowire width.

As shown in Figure 4a, when the width of the gold nanowires is 172 nm (*λ*_1_ = 774 nm), 176 nm (*λ*_2_ = 792 nm), and 180 nm (*λ*_3_ = 810 nm), the reconstructed spectral information at *λ*_1_, *λ*_2_, *λ*_3_ are missing. Therefore, the gold nanowire width should not equal to an integer multiple of the stationary wave period. Considering the current micro-nano processing technology [31,35], when the gold nanowire width is 100 nm, the reconstructed spectral information is shown in Figure 4b (red line), and a local amplification is shown in Figure 4c. The corresponding normalized intensity maximum error of the reconstructed spectrum is 3.87%.

### 3.3. Raman Spectroscopy Detection

In order to see the performance of this method, an experimental Raman spectrum of R6G molecule from SERS (surface enhanced Raman scattering) substrate at an excitation light wavelength of 532 nm is used as the original spectral information. The Raman shift from 400 cm^−1^ to 2000 cm^−1^ corresponds to the wavelength range from 543.57 nm to 595.34 nm. When the distance from the initial detection position is 70 μm away from OPD = 0, it is almost impossible to record any static fringe data. Therefore, we set ∆*L* = 667 μm, while a dynamic peak position distance 332.5 μm away from OPD = 0, and we start to detect at 125 μm away from OPD = 0. According to the resolution *R* = 1/(2*n_s_z_max_)*, we set *z_max_* = 500 μm in order to achieve a resolution of 5 cm^−1^. Based on the Nyquist-Shannon criterion and ∆σ = 1/(4*n*∆*z*), the sampling interval should be less than 59.1 nm at 532 nm, so the reconstructed spectral bandwidth ∆σ is larger than 1.88 × 10^4^ cm^−1^. When only one gold nanowire is used, the required number of moving steps is *z_max_*/∆*z*. Taking consideration of a 7 μm pixel size of the traditional detector, we set ∆*z* = 40 nm, and 25 gold nanowires are asymmetrically placed on the *L*_5_. The positions of the first nanowire and the No.N nanowire on the left are 145 μm and 145 + 40(N−1) μm away from OPD = 0, and the positions of the first nanowire and the No.N nanowire on the right are 165 μm and 165 + 40(N−1) μm away from OPD = 0. The corresponding sampling data of the 1st, 6th, 11th, 16th and 21st nanowires and the position of the gold nanowires corresponding to the dynamic interferogram are shown in Figure 5. Based on Equation (9) and ∆*z_central_* ≈ 2.6667 × 10^−3^∆*n*, the temperature change is 0.242 K when sampling interval is 40 nm, the required temperature range is 121 K and the required temperature tuning times is 500.

Original Raman intensity information is shown in Figure 6a. The ideal reconstructed results are shown in Figure 6b when not considering the gold nanowires width. The reconstructed spectrum (red) and the original spectra (black) near the typical characteristic peaks of 1365 cm^−1^ and 1509 cm^−1^ are shown in Figure 6c. The results show that the reconstructed spectrum is very consistent with the original spectrum.

In the case of an incident wavelength of 532 nm, Raman peaks of 1365 cm^−1^ and 1509 cm^−1^ correspond to the wavelength of 573.7 nm and 578.4 nm, respectively. The generated stationary waves period has a range from 127.5 nm to 128.5 nm. Setting the gold nanowires width to 100 nm, 127.5 nm, 128.5nm, 150 nm and 175 nm, respectively, the reconstructed results are shown in Figure 7a,b. We find that the signal near the wavelength of 573.7 nm is missing (red line) when the gold nanowire width is 127.5 nm; the signal near the wavelength of 578.4 nm is missing (blue line) when the gold nanowire width is 128.5 nm, so we conclude that the gold nanowire width should not be equal to an integer multiple of the stationary wave period.

The results also show that when the gold nanowires width is 100, 150 and 175 nm, the corresponding position error at the Raman peak of 1365 cm^−1^ and 1509 cm^−1^ is 0.774 cm^−1^ and 2.574 cm^−1^, respectively. The corresponding normalized intensity errors of the reconstructed spectrum are 37.17%, 1.92% and 19.62% at 1365 cm^−1^, 37.07%, 5.52% and 14.04% at 1509 cm^−1^, respectively. Taking into consideration the current processing technologies, the gold nanowire width of 150 nm could be more cost-effective.

## 4. Conclusions

An all-on-chip Si_3_N_4_ waveguide DTM-SWIFT system was proposed and analyzed in detail. The results of our theoretical analysis show that the method can complete a resolution of 6 nm for Gauss spectrum reconstruction using only six gold nanowires when the temperature tuning range is less than 24.2 K and the temperature tuning times are less than 100, and a resolution of 5 cm^−1^ for Raman spectrum reconstruction using only 25 gold nanowires when the temperature tuning range is less than 121 K and the temperature tuning times are less than 500. The reconstructed Raman spectrum can achieve a peak position error of less than 1 cm^−1^ at the Raman peak of 1365 cm^−1^. This idea is expected to be combined with a waveguide Raman sensor to realize an on-chip Raman spectrum sensing and detection system.

## Figures and Tables

**Figure 1 sensors-21-02352-f001:**
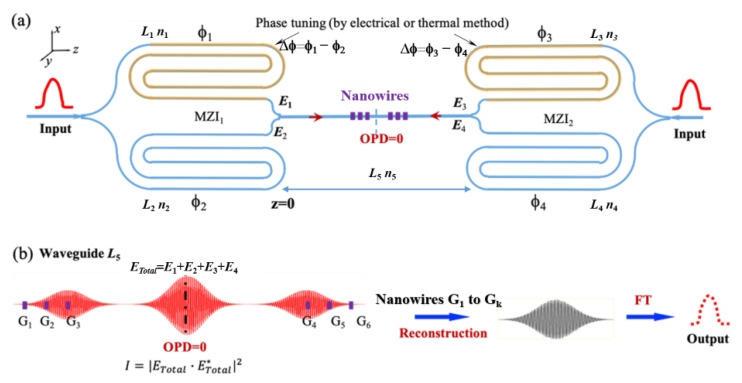
(**a**) Schematic diagram of the DTM-SWIFT system. (**b**) Sampling and reconstruction process. The upper arm lengths of MZI_1_ and MZI_2_ are *L*_1_ and *L*_3_, and the lower arm lengths are *L*_2_ = *L*_1_
*+* ∆*L* and *L*_4_ = *L*_3_
*+* ∆*L*, and the corresponding refractive index is *n*_1_, *n*_2_, *n*_3_ and *n*_4_. The length of the straight waveguide in the middle is *L*_5_, and the corresponding refractive index is *n*_5_. ∆*L* is an initial arm length difference between two arms of each MZI, which is determined by an initial phase.

**Figure 2 sensors-21-02352-f002:**
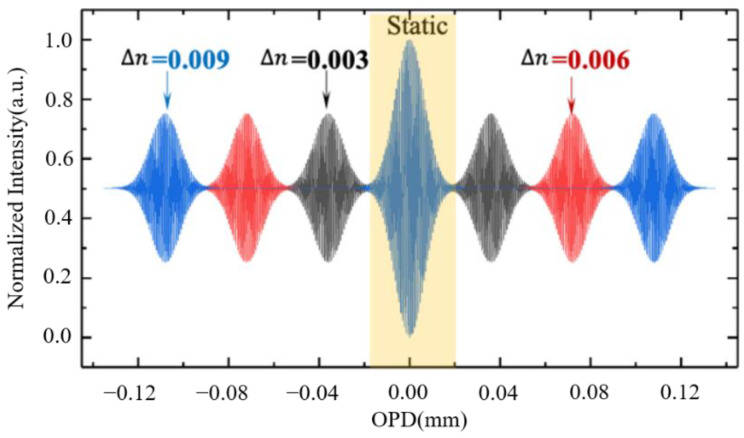
The normalized distribution of the optical signals on the straight waveguide *L*_5_, the symmetrical distributions on both sides of OPD = 0 correspond to ∆*n* = 0.003 (black), 0.006 (red), and 0.009 (blue).

**Figure 3 sensors-21-02352-f003:**
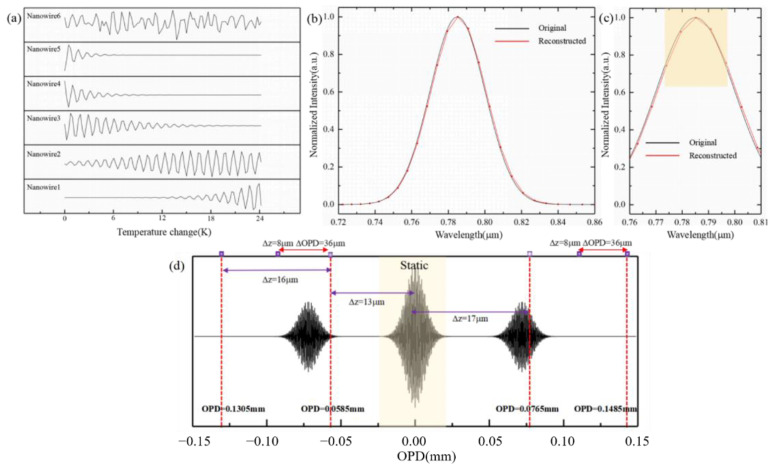
(**a**) Sampling data recorded by 6 nanowires at each tunable temperature, (**b**) the reconstructed spectrum (black) obtained by FT and the original incident Gauss spectrum (red), (**c**) a partial enlarged view, (**d**) the position of the gold nanowires corresponding to the dynamic interferogram.

**Figure 4 sensors-21-02352-f004:**
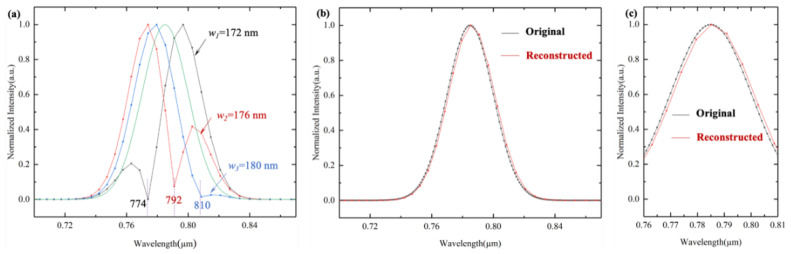
(**a**) Reconstructed spectral information under different nanowire widths. (**b**) Reconstructed spectral information while the nanowire width of 100 nm and the original spectrum, (**c**) local amplification in (**b**).

**Figure 5 sensors-21-02352-f005:**
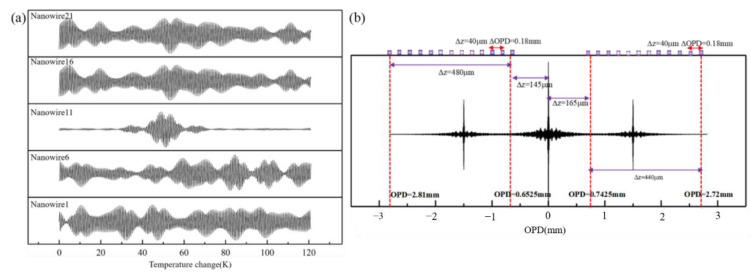
(**a**) Raman spectroscopy detection process. The corresponding sampling data of the 1st, 6th, 11th, 16th and 21st nanowires. (**b**) The position of the gold nanowires corresponding to the dynamic interferogram.

**Figure 6 sensors-21-02352-f006:**
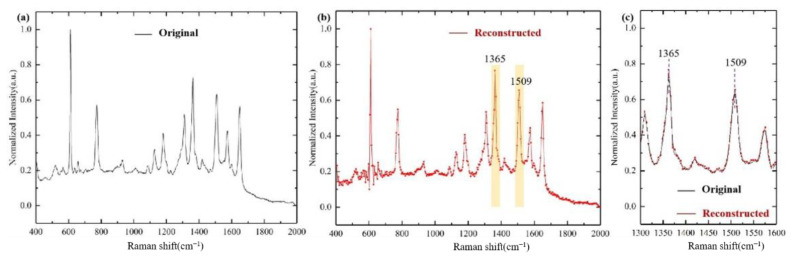
(**a**) Original Raman intensity information; (**b**) Reconstructed Raman information; (**c**) Typical characteristic peaks near 1365cm^−1^ and 1509cm^−1^ restored spectrum (red) and original spectrum (black).

**Figure 7 sensors-21-02352-f007:**
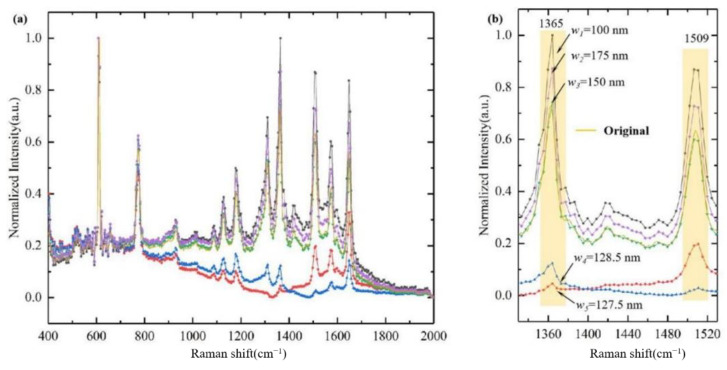
(**a**) The original Raman intensity (yellow) and reconstructed information when the gold nanowires width is 100 nm (black), 127.5 nm (red), 128.5 nm (blue), 150 nm (green) and 175 nm (purple); (**b**) enlarged information at typical characteristic peaks of 1365 cm^−1^ and 1509 cm^−1^.

## Data Availability

Not applicable.

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
