# Peer review of "Dual Tunable MZIs Stationary-Wave Integrated Fourier Transform Spectrum Detection"

_sensors, 2021, doi:10.3390/s21072352_

Round 1
Reviewer 1 Report
This manuscript reports on the development of an on-chip Si3N4 waveguide stationary-wave integrated Fourier transform spectrum platform using double tunable mach-zehnder interferometers. Their extensive studies demonstrated that the proposed approach provide a resolution of 6 nm for Gauss spectrum reconstruction using only 6 gold nanowires when temperature tuning range less than 24.2 K, and a resolution of 5 cm^-1 for Raman spectrum reconstruction through the use of only 25 gold nanowires when temperature tuning range less than 121 K. Although the work is well-supported with plenty of examinations and assessments, it suffers from important shortcomings that listed below. I suggest the authors to carefully address the comments and revise their work based on mentioned concerns.
General comments:
1) Most of the employed references of the bibliography are out of date. The authors have to consider the recent advances in the field and provide practical examples based on a mixture of initially proposed and modern technologies.
2) Although the authors considered and exemplified most of available on-chip spectrometers, they missed one of the most promising and important types of these devices. Plasmonic spectrometers are unique devices that are missed in this work. see Tsur et al. Optics Letters 41(15), 3523-3526 (2016).
3) Some additional examples could be provided for the use of both dielectric and plasmonic waveguiding platforms in diverse telecommunication applications, from optical power splitters to demultiplexers, and interferometers. see Liu et al. Khalilou et al. Optics Communications 321, 56-60 (2014), ACS Nano 6, 5482-5488 (2012), Ye et al. IEEE Transactions on Nanotechnology, 18, 617-62 (2019), and many other examples.
4) All acronyms must be explained in complete format before using. For instance, MZI.
5) Conclusions is too short and must be extended.
Technical comments:
1) In connection to the second comment of the general comments section above, the authors should compare the advantages of the developed spectrometer with plasmonic or hybrid tools in the literature.
2) What is the influence of losses within gold nanowire arrays on both sides of OPD?
3) The decay length of the waveguide must be quantified and the influence of the studied parameters on that should be argued.
4) How the theoretical part of the work was accomplished? Through numerical analysis or MATLAB Coding? or Something else? This must be explained by providing a separate subsection for this topic.
Reviewer 2 Report
Summary:
This manuscript described a theoretical analysis of a dual tunable MZI spectrum detection method with stationary-wave integrated Fourier Transform. Based on theoretical analysis, the proposed spectrum detection method can achieve a spectrum resolution of 6 nm with 6 gold nanowires and 5 cm^-1 for Ram spectrum reconstruction with 25 gold nanowires. The manuscript is well-written, although the final claim about the resolution seems a little bit ambiguity and need to be emphasized by the authors that it is based on theoretical analysis only.
Major Review:
1. Can the authors address with more clear mannar in the conclusion (both in abstract and end of the paper) that the results of this manuscript is based on theoretical analysis instead of experiment? Otherwise, the claim of the results is a little bit misleading to the audience.
2. Page 6:
The description of the positions of the 25 nanowires for Raman spectroscopy detection can be improved in the manuscript.
=> Can the authors also draw a graph or cartoon to demonstrate the positions of the 25 nanowires for the Raman spectroscopy detection? It is also fine to be similar to Fig3 (c).
3. Page 7:
How is the temperature influence the the Raman spectroscopy detection? Can the author provide some sampling data recorded by nanowires at each tunable temperature? It can be similar to Fig3(a).
Minor Review:
1. Page 1:
MZI
=> What is MZI stands for? Can the author provide the full name of MZI for the time in the manuscript?
2. Page 5:
Figure 3 (a)
=> Can the author slightly increase the fonts in fig3(a), e.g. for "Groove X"
Reviewer 3 Report
This manuscript presents an all-on-chip spectrum detection system based on Si3N4 waveguide dual tunable MZIs stationary-wave integrated Fourier transform technology (DTM-SWIFT). By applying gold nanowires to sample the interference fringes, one can reconstruct the Gauss spectrum as well as the Raman spectrum with relatively high resolution. The authors combined a systematic theoretical analysis and discuss the key influencing factors, and also compare with the experimental results to demonstrate the feasibility of on-chip spectrometers. In my option, this is a good piece of work with some simplicity, novelty, and practicability, however, various issues listed below have hindered the publication of the article in the present form and should be addressed.
1. In the introduction part, the authors should describe why selecting Si3N4 waveguide.
2. Have the authors considered the sensitivity to temperature variations and limitations of the designed technologies?
3. Can the authors explain why do they choose 532 nm instead of 405 or 785 nm as the Raman excitation wavelength to acquire Raman spectrum of R6G for comparison?
4. In section 3.3, could the authors explain how did they calculate the normalized intensity errors?
Small points:
1. Write out the acronym prior to first use, i.e. MZI (page1, title, and line 11) CCD (page 2, line 45), and SNOM (page 4, line 121).
2. There are some small errors along with the manuscript. For example, Si3N4 not SI3N4 in equation (6), line 109, and equation (7) on page 3. Please check carefully.
Following the above considerations, I suggest a minor revision before acceptance of the manuscript.
Round 2
Reviewer 1 Report
The comments and concerns have been responded correctly, hence, the work is publishable in the current format.
Reviewer 2 Report
No further comments